# Long-Term Outcomes of Breast Cancer Patients Receiving Levobupivacaine Wound Infiltration or Diclofenac for Postoperative Pain Relief

**DOI:** 10.3390/pharmaceutics15092183

**Published:** 2023-08-23

**Authors:** Josipa Glavas Tahtler, Dajana Djapic, Marina Neferanovic, Jelena Miletic, Marta Milosevic, Kristina Kralik, Nenad Neskovic, Ilijan Tomas, Dora Mesaric, Ksenija Marjanovic, Jasmina Rajc, Zelimir Orkic, Ana Cicvaric, Slavica Kvolik

**Affiliations:** 1Department of Anesthesiology, Resuscitation and Intensive Care, Osijek University Hospital, 31000 Osijek, Croatia; josipa.glavastahtler@kbco.hr (J.G.T.); ana.cicvaric@kbco.hr (A.C.); 2Faculty of Medicine Osijek, Josip Juraj Strossmayer University of Osijek, 31000 Osijek, Croatia; maneferanovic@mefos.hr (M.N.); jmiletic@mefos.hr (J.M.); mmilosevic@mefos.hr (M.M.); kristina.kralik@mefos.hr (K.K.); ilijan.tomas@kbco.hr (I.T.); dora.mesaric@kbco.hr (D.M.); ksenija.marjanovic@kbco.hr (K.M.); jasmina.rajc@kbco.hr (J.R.); zelimir.orkic@kbco.hr (Z.O.); 3Department of Oncology and Radiotherapy, Osijek University Hospital, 31000 Osijek, Croatia; 4Department of Pathology and Forensic Medicine, Osijek University Hospital, 31000 Osijek, Croatia; 5Department of Surgery, Osijek University Hospital, 31000 Osijek, Croatia

**Keywords:** breast cancer, acute postoperative pain, postoperative analgesia, patient-controlled analgesia, levobupivacaine, diclofenac, hand grip strength, quality of life, treatment outcome

## Abstract

Breast cancer is the most common malignant disease in women. Preclinical studies have confirmed that the local anesthetic levobupivacaine has a cytotoxic effect on breast cancer cells. We examined whether postoperative wound infiltration with levobupivacaine influences survival in 120 patients who were operated on for breast cancer and underwent quadrantectomy or mastectomy with axillary lymph node dissection. Groups with continuous levobupivacaine wound infiltration, bolus wound infiltration, and diclofenac analgesia were compared. Long-term outcomes examined were quality of life, shoulder disability, and hand grip strength (HGS) after one year and survival after 5 and 10 years. Groups that had infiltration analgesia had better shoulder function compared to diclofenac after one year. The levobupivacaine PCA group had the best-preserved HGS after 1 year (*P* = 0.022). The most significant predictor of the 5-year outcome was HGS (*P* = 0.03). Survival at 10 years was 85%, 92%, and 77% in the diclofenac, levobupivacaine bolus, and levobupivacaine PCA groups (ns. *P* = 0.36). The extent of the disease at the time of surgery is the most important predictor of long-term survival (*P* = 0.03). A larger prospective clinical study could better confirm the effect of levobupivacaine wound infiltration on outcomes after breast cancer surgery observed in this pilot study—trial number NCT05829707.

## 1. Introduction 

Breast cancer is the most common cancer in women worldwide. Surgical removal is an important way of treating breast cancer [1]. Depending on the characteristics and extent of the tumor, the type of surgery is selected—lumpectomy, quadrantectomy, or mastectomy. The involvement of the axillary lymph nodes determines whether the operative procedure will include sentinel lymph node biopsy (SLNB) or axillary lymph node dissection (ALND). Pronounced acute postoperative pain after breast surgery is present in half of the operated patients and is associated with ALND [2]. If acute pain is not adequately treated, it can lead to chronic pain with limited shoulder movement [3]. In addition to acute postoperative pain, postmastectomy pain syndrome (PMPS) is also an important problem. PMPS is neuropathic pain of the chest wall, axilla, or arm, which can occur after any breast surgery and lasts at least 6 months [2]. The origin of PMPS is unknown, but it is considered that the intensity of acute pain is proportional to the appearance of chronic pain [2]. Axillary evacuation, radiotherapy, chemotherapy, and psychological factors can contribute to the development of PMPS [4]. The quality of life is impaired in patients with PMPS, and it influences the psychosocial and physical performance of patients operated on for breast cancer. Improving postoperative analgesia is the main step in the prevention of PMPS as well as improving the quality of life after surgery [2]. In addition to the choice of analgesic, the method of drug administration for postoperative analgesia may be related to the outcomes. Patients can have a greater sense of satisfaction if they themselves control the amount of medication for postoperative analgesia [5]. This method of drug delivery is known as patient-controlled analgesia (PCA).

Local anesthetics are useful drugs for intraoperative and postoperative analgesia after breast cancer surgery [6]. Levobupivacaine, the s-isomer of bupivacaine, is widely used in anesthesia for single-shot nerve blocks and neuraxial anesthesia. Due to its lower toxicity, it is also suitable for continuous postoperative analgesia. In addition to the analgesic effect, it has been shown that levobupivacaine has an antiproliferative effect [7].

Li et al. confirmed on breast cancer cell lines that levobupivacaine has a cytotoxic effect on MDA-MB-231 and MCF7 breast cancer cells [7]. This effect is dose-dependent. After exposure of breast cancer cells to levobupivacaine in a higher concentration for 48 h, their viability and migration decrease by >40%. At the same time, it did not affect non-tumor breast cells [7]. Levobupivacaine had the same effect on triple-positive BT-474 breast cancer cells [8]. The mechanism involved in the antiproliferative effect is the inhibition of the cell cycle process from phase S to phase G2/M [7,9] and the induction of apoptosis of breast cancer cells by suppressing signaling through the PI3K/Akt/mTOR pathway [10].

The extent of breast resection is one of the factors associated with DFS and local recurrences [1]. Pain in the shoulder and weakness of the arm on the operated side are possible complications of breast cancer surgery [11]. Postoperative complications, as well as the type of anesthesia, can affect treatment outcomes [2,12]. Retrospective analyses have shown that the use of lidocaine in infiltration analgesia during cancer surgery is associated with improved survival [13]. Although levobupivacaine wound infiltration could be associated with better long-term outcomes, there are no clinical studies on the effect of levobupivacaine on survival so far.

Diclofenac is an NSAID that has proven to be an effective analgesic for analgesia after breast cancer surgery [14]. Combined with paracetamol, it reduces the use of opioids but leads to greater blood loss compared to a placebo [14]. To date, there are no studies examining long-term survival in patients receiving diclofenac or local anesthetics for postoperative analgesia.

This study aims to examine the differences in the long-term outcome of patients who, after quadrantectomy or mastectomy, received wound infiltration with levobupivacaine or diclofenac for postoperative analgesia. Health-related quality of life (HRQOL), hand grip strength, and shoulder disability on the operated side will be analyzed after 1 year, and long-term survival after 5 and 10 years.

## 2. Materials and Methods

Out of a total of 149 patients who underwent surgery for breast cancer in the period 2009–2012, 120 of them were included in the prospective randomized parallel study. By drawing numbers from a hat, the patients were randomized into groups as follows 1–50 controls, 51–100—levobupivacaine bolus, and 101–150 levobupivacaine PCA. The study included female breast cancer patients 30–75 years old, ASA 1–3, undergoing quadrantectomy or mastectomy with ALND. The patients who refused to sign informed consent, psychiatric patients, those who had allergies to local anesthetics or diclofenac, patients unable to read and write, and women who previously had an operation on the armpit of the operated side were not included in the study. All patients received the same explanation: that they will receive postoperative analgesia and that the aim of the study is to see if this analgesia is satisfactory for postoperative pain control. They were unaware of other methods of analgesia that were used in other groups. The study was approved by the decision of the ethics commission KBC Osijek 29-1:1939-10/2008, trial number NCT05829707. Demographic data were recorded for all patients before surgery: age, body mass index, ASA status, and muscle strength measured by hand grip strength. All patients were assessed for quality of life using the Croatian version of the SF-36 questionnaire before surgery [15]. Eight domains were examined: physical health, role limitation due to physical problems, pain, general health perception, energy/vitality, social functioning, limitations due to emotional problems, and general mental health. Higher sum values represent better health status [15]. The pain assessed by the SF-36 questionnaire (bodily pain) also referred to other locations unrelated to surgery, such as headache, low back pain, or arthritis. Shoulder pain was assessed using the Shoulder Disability Questionnaire (SDQ) in 16 typical situations, such as writing, opening a door, sleeping on the operated side, or carrying a load [16]. Placing a catheter for analgesia in the axilla may lead to tingling and hand weakness, in addition to analgesia of the area where ALND was performed. Therefore hand grip strength (HGS) was measured bilaterally for comparison with the non-operated side and objective measurement of postoperative weakness. Hand grip strength was measured preoperatively, 4 days after surgery and at follow-up after 1 year, using a handheld dynamometer (Dynatest^®^, Rud. Reister Gmbh&CaKG, Jungingen, Germany) and expressed in bars. Each measurement was presented as an average of three consecutive measurements.

### 2.1. Postoperative Analgesia

Before the operation, the patients were randomized into 3 groups: the first group received diclofenac for postoperative analgesia. On the first day, they received 2 × 75 mg intravenously and then 3 × 50 mg tablets. For the wound infiltration with levobupivacaine, the surgeon placed a perforated catheter at the end of the surgical procedure in both groups of patients. The tip of the catheter was in the axillary fossa, where the dissection was performed. Patients in the levobupivacaine bolus group (N = 39) received three times a day bolus doses of 0.5 mg/kg 0.5% levobupivacaine (Chirocaine, Abbot S.P.A., Latina, Italy) through a catheter with an antibacterial filter. The dose of levobupivacaine was prescribed by the doctor, and the drug was delivered by the nurse. In the levobupivacaine PCA group (N = 40), women had the same catheter placed but received 0.05 mg/kg/h 0.5% levobupivacaine continuously for 24 h via a catheter placed in the axilla. Levobupivacaine was delivered by PCA pump (CADD—Legacy^®^ PCA Pump, Model 6300, Smiths Medical MD, Inc., St. San Diego, CA, USA). These patients were allowed to add a dose of 7.5 mg 0.5% levobupivacaine in case of pain by pressing the patient’s button on the PCA pump, with a lockout period of 4 h. If the patients in any wound infiltration group felt weakness or paresthesias in their hands, the drug dose was reduced to 75% of the calculated dose. Postoperative pain was measured from 1–4 postoperative days at rest and movement using the numerical rating scale (NRS), where 0 means that they feel no pain at all and 10 is the worst possible pain. All patients had the option of additional analgesia, using NSAIDs for pain < 4 or weak opioids for NRS ≥ 4, delivered by a nurse on request. After the surgery, the patients received oncological chemoradiotherapy, depending on the type and stage of the cancer. 

### 2.2. Long-Term Outcomes

The outcome of the patients was assessed at the surgical follow-up examination 1 year after the surgery. All patients completed the SF-36 and SDQ questionnaire, and their HGS was measured. In February 2023, after the approval of the Ethics Committee (R1-1510/2023) was obtained, the extent of the cancer at the time of surgery and the current status of the disease was checked from the hospital records. The stage of the disease was scored postoperatively according to the American Joint Committee on Cancer Breast Cancer Staging System from 1 for in situ cancer to 9 for metastatic cancer [17]. Metastatic disease was confirmed at postoperative control based on histological findings, tumor markers, and imaging techniques where indicated. For the patients not found in the hospital registry, we checked whether they were alive and the date of death for those who died through the official population records and death registers of the Republic of Croatia.

### 2.3. Statistical Analysis 

Categorical data are represented by absolute and relative frequencies. The normality of the distribution of numerical variables was tested with the Shapiro–Wilk test. Data are presented as median (interquartile range, IQR). The Kruskal–Wallis test (post hoc Conover) was used to test differences of continuous variables between three or more independent groups. Kaplan–Meier survival curves were compared using the log-rank test. Differences of continuous variables between two measurements were tested with Wilcoxon’s test and between multiple measurements with Friedman’s test or the marginal homogeneity test. Logistic regression (bivariate and multivariate) was used to examine which predictors were significant in predicting 5 and 10-year survival.

All *p* values are two-sided. The significance level was set at alpha = 0.05. The minimum required sample size with a significance level of 0.05 and a power of 0.8 to detect an effect f = 0.287 in the difference of numerical variables between three independent groups of subjects (diclofenac, levobupivacaine bolus, levobupivacaine PCA), was 120 subjects. The statistical programs used were MedCalc^®^ Statistical Software version 20.218 (MedCalc Software Ltd., Ostend, Belgium; https://www.medcalc.org, (accessed on 20 January 2023); 2023) and SPSS ver. 23 (IBM Corp. Released 2015. IBM SPSS, Ver. 23.0. Armonk, NY: IBM Corp.).

## 3. Results

A total of 149 patients signed their consent to participate in the study. After patients who refused further participation and control examination (n = 5), who unintentionally removed wound infiltration catheters (N = 2), who after histological analysis did not require axillary lymph node dissection (N = 21), and 1 due to diclofenac intolerance were excluded, 120 patients were analyzed in the study (Appendix A). Their demographic data are presented in Table 1. There was no difference in any of the parameters recorded between the groups.

Postoperative pain in rest and movement was well controlled in three groups with a median NRS ≤ 3 (Appendix A). A significant reduction in pain was registered from the first to the fourth postoperative day (Friedman’s test, *P* < 0.001). The lowest pain was observed in movement in the levobupivacaine PCA group on the third day (Kruskal–Wallis’s test, *P* = 0.043), as shown in Appendix A.

HRQOL before surgery and one year after is shown in Figure 1a,b. In the preoperative perception, there were no statistically significant differences between the groups in any domain. Analyzing the change in perception within the same group, a statistically significant perception of impaired physical health was recorded in the groups with levobupivacaine, especially in the levobupivacaine PCA group (*P* = 0.003). In the levobupivacaine bolus group, a somewhat higher perception of bodily pain compared to preoperative values (*P* = 0.042) and somewhat better social functioning in the PCA analgesia group (*P* = 0.046) was recorded one year after surgery (Figure 1b). No statistically significant changes in role limitation due to physical problems or role limitation due to emotional problems were recorded. Other health domains, such as energy vitality, general health perception, or general mental health, did not change significantly in the patients during the one-year period (Appendix A).

Shoulder disability was measured after one year in each group and compared to the preoperative condition. In the diclofenac group, there was a statistically significant worsening of shoulder function in 4 out of 16 common situations (*P* < 0.05), in levobupivacaine bolus in one situation (*P* = 0.01), and in PCA in none of the situations evaluated by SDQ (Appendix A).

Measurement of hand grip strength preoperatively did not confirm differences between groups. The levobupivacaine PCA group had significantly better postoperative HGS on the operated side (Friedman’s test *P* = 0.022), while in other groups, HGS was somewhat weaker during the follow-up period (Appendix A).

The impact of analgesia on long-term survival was evaluated after 5 and 10 years. Cox bivariate regression analysis confirmed that HGS was the only factor significantly predicting 5-year survival (HR = 0.001). The same was confirmed by multivariate regression analysis (Table 2).

There were certain differences in survival between the groups, e.g., 10-year survival was 92% in the levobupivacaine bolus group compared to 77% in the PCA group, but none of the indicators was statistically significantly associated with 10-year survival (Appendix A).

Overall survival at the end of the study, after 159 months, was still better in the group that received bolus levobupivacaine (Figure 2a), but it was not statistically significant. There was also a difference in survival associated with ASA status, although this was again not statistically significant (Figure 2b, Appendix A).

The current status of the disease was visible for most of these patients through the hospital information system due to controls after surgery or treatment of other diseases. By the time of the analysis, after a mean of 159 months, 23 (19.2%) patients had died. Active disease and current oncological therapy were recorded in 10 patients, and 87 of them were without signs of breast cancer. Age at diagnosis of breast cancer (*p* = 0.03, HR 1.05, 95% CI 1.03 to 1.11) and tumor stage at the time of surgery are significantly associated with long-term outcomes and current disease status (Table 3).

## 4. Discussion

In this study, we examined whether methods of drug delivery for postoperative analgesia and not only the type of drug were associated with long-term outcomes. After 1 year, we analyzed shoulder disability, HGS, and HRQOL. Measurement of shoulder disability confirmed that pain was reduced in both groups that received levobupivacaine wound infiltration compared to the preoperative. In the diclofenac group, shoulder disability was increased in several typical situations, such as carrying loads and raising the arm.

Although HGS decreased after 1 year, this decrease was the least in the levobupivacaine PCA group. Hand grip strength was confirmed as a significant predictor of 5-year survival in our study. This has also been observed in other studies. The correlation between HGS and frailty was observed by Meerkerk and colleagues after neck cancer operations [18]. HGS has also been shown to be a significant predictor of early poor outcomes in cancer patients, including those undergoing chemoradiotherapy [19].

Shoulder pain on the operated side was stronger in all groups after 1 year. In the group with diclofenac analgesia, it was significantly higher in six typical movements, such as carrying a load, raising the arm, or opening the door. In the group with infiltration analgesia, which reported the severest bodily pain, the pain in the shoulder was significantly stronger only when lying on the operated shoulder. It is important to say that no group had role limitations due to physical or emotional problems. Reasons for persistent postoperative shoulder pain, apart from the application of analgesia, may be the occurrence of postoperative inflammatory changes, which later lead to increased scar formation. In patients undergoing axillary lymph node dissection, this can result in reduced shoulder mobility, lymphedema, and persistent chronic pain [20]. We assume that local vasodilation is the possible reason for better shoulder mobility in patients who received levobupivacaine for infiltration analgesia. This can result in better blood circulation and reduced local scar formation surrounding the tip of the catheter in the axilla. We, therefore, believe that decreased shoulder disability observed in the levobupivacaine groups can support the choice of postoperative analgesia by wound infiltration with local anesthetics. 

Breast cancer treatment, which includes surgery, chemotherapy, and radiation, can lead to significant deterioration in several domains of quality of life, including pain, role function, physical function, and cognitive function [20]. Shoulder pain, which existed to a certain degree in most women before surgery, intensified in all groups of our patients after breast cancer surgery. Reduced shoulder and upper arm mobility and persistent shoulder pain are observed in 30–50%, and lymphedema in 15–25% of women after breast cancer surgery [20]. According to a study by Klein et al., shoulder pain and reduced range of motion after breast cancer surgery were significant. They confirmed that shoulder pain is associated with stronger acute postoperative pain, tumor size, and the number of dissected lymph nodes [11]. According to the average age, most of our patients are postmenopausal, so the frequency of joint pain is expected to be higher than in younger women. An interesting finding from the study by Longo and coworkers from 2021 is that the presence of estrogen receptors α and β was found in the supraspinatus tendons and that the number of these receptors decreases with menopause in women [21]. This observation may explain why shoulder pain is common in postmenopausal women, as well as the fact that it worsens after breast cancer therapy with antiestrogens [22]. In this context, in a certain number of breast cancer patients, local anesthetic infiltration could be useful and may reduce the progression of chronic pain.

With this size of the groups, there were differences between the groups in the outcomes measured after 1 year and a trend of better survival of patients who received bolus levobupivacaine. We expected that, due to the toxicity of levobupivacaine on cancer cells, these groups would have reduced migration of cancer cells postoperatively and better survival. Bolus application of local anesthetic for infiltration of the operative area achieves higher concentrations of local anesthetic in the surgical wound and systemic circulation than continuous administration via the PCA pump. It is possible that in this way, more significant toxicity of the local anesthetic is achieved on cancer cells. It reduces their viability, migration, disintegration, and consequently, the occurrence of local and distant metastases. This is a possible explanation for the differences in survival between the two groups, which did not reach statistical significance.

The average age at the onset of breast cancer in our population is 56 years, while it was 48 in Arabs, 50–54 in China, and 55–59 in the European Union and the United States [23,24]. In addition to the patient’s age, comorbidities, and perioperative procedures, the survival of patients with breast cancer is influenced by socioeconomic factors, such as the level of education and the availability of health care [25]. Better physical condition and exercise can contribute to a better quality of life in patients after breast cancer surgery [26,27,28]. We, therefore, expected that patients who had better HGS, as well as those in the group with bolus administration of levobupivacaine, could have better long-term survival. Although HGS was the best predictor of 5-year survival, in the bivariate regression, more significant predictors of 10-year survival were the age at diagnosis and the stage of the disease at diagnosis. High BMI was associated with an increased risk of breast cancer and worse survival in several studies [29,30], but in our patients, this effect was not statistically significant.

Long-term survival in this study corresponds to that observed in other European countries [31]. It was 85% in all the patients after 10 years, and it was slightly better, although not statistically significant, in the group with bolus analgesia with 0.5% levobupivacaine. In countries with a lower standard, survival is lower, such as that reported in Iran, where 10-year survival was 69% [32].

The shortcoming of this study is the small number of subjects included in it and non-randomization. Considering the non-standard tests and interventions carried out during the initial hospitalization and follow-up after one year, it is difficult to include a large number of patients. In addition to the factors monitored here, the ongoing COVID-19 pandemic most likely increased mortality in this population, as observed in other studies [33]. Considering that data such as those about the COVID-19 infection are not visible in the hospital IT system, they were not considered when processing this data. Cancer type was also not included in the analysis because it was not known when randomization was performed [34]. The data on the spread of the disease at the time of the analysis also does not have to be completely accurate. The evaluation of the extended disease at the time of the analysis was conducted only on the basis of the implementation of the therapy or the records at the last examination by the oncologist and not on the basis of objective indicators, such as tumor markers.

It cannot be claimed that the antiproliferative effects observed with levobupivacaine are drug-specific. It is possible that such effects are also characteristic of other local anesthetics [35]. In an in vitro study, bupivacaine reduced the size and increased the number of apoptosis by acting on ERK1/2 and STAT3 signaling [36]. A similar effect on postoperative pulmonary metastasis in a murine model was observed with the use of lidocaine by its inhibitory effect on MMP-2 expression [37]. The possible mechanisms of these antiproliferative effects may be an effect on caspase-3 activation, a decrease in BCL-2 protein, and suppression of PI3K/Akt and MAPK signaling pathways [38,39]. Therefore, this “side effect” of local anesthetics is certainly desirable and may be another reason for their perioperative use. Considering the observed dose-dependent effect, as is the case in our study, their local infiltration near the site of tumor resection will certainly achieve the best effect on analgesia and possible inhibition of migration.

Since this study was completed, changes have been made in breast cancer treatment in our institution and worldwide; as such, the study could not be repeated in the same way. The best treatment approach is decided at weekly meetings by a multidisciplinary team of specialists, including a plastic surgeon, oncologist, pathologist, and radiologist. The multidisciplinary team may prescribe neoadjuvant chemotherapy according to the tumor size, predictive biomarkers, and axillary lymph node status. Protocols are regularly adjusted to comply with international standards of care. The benefit of the neoadjuvant approach is the possibility of rendering inoperable tumors operable and facilitating breast conservation by tumor down-staging. It also offers valuable prognostic information about treatment response.

With such treatment before surgery, the migration potential and, thus, tumor metastasis in the perioperative period may be reduced. For some patients who do not undergo neoadjuvant chemotherapy or in whom lymph node dissection is performed, infiltration of the wound with levobupivacaine still presents the possibility of achieving good postoperative analgesia, preserved hand function, and reduced metastatic potential. The long-term outcomes of breast cancer patients with such personalized treatment could also be better.

## 5. Conclusions

This pilot study confirmed that postoperative wound infiltration analgesia with levobupivacaine had a beneficial effect on the preservation of shoulder function on the operated side in patients after breast cancer surgery with axillary dissection. For some long-term outcomes, not only the type of drug used for postoperative analgesia was important, but also the mode of drug delivery. No statistically significant difference in long-term survival was observed in patients who had postoperative analgesia with levobupivacaine delivered through a wound infiltration catheter and those who had analgesia with diclofenac. A prospective randomized study, which would evaluate a larger number of patients and data, could provide better data on the real impact of levobupivacaine on survival.

## Figures and Tables

**Figure 1 pharmaceutics-15-02183-f001:**
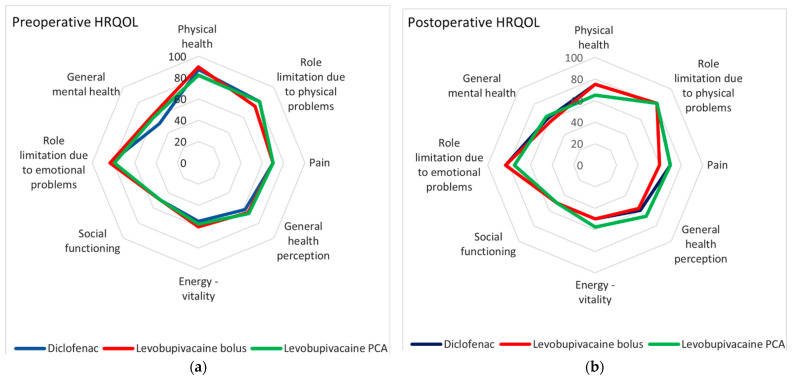
Health-related quality of life (HRQOL) in breast cancer patients undergoing quadrantectomy or mastectomy, who received wound infiltration with levobupivacaine bolus, levobupivacaine patient-controlled analgesia (PCA), or diclofenac for postoperative pain relief before (**a**) and 1 year after surgery (**b**).

**Figure 2 pharmaceutics-15-02183-f002:**
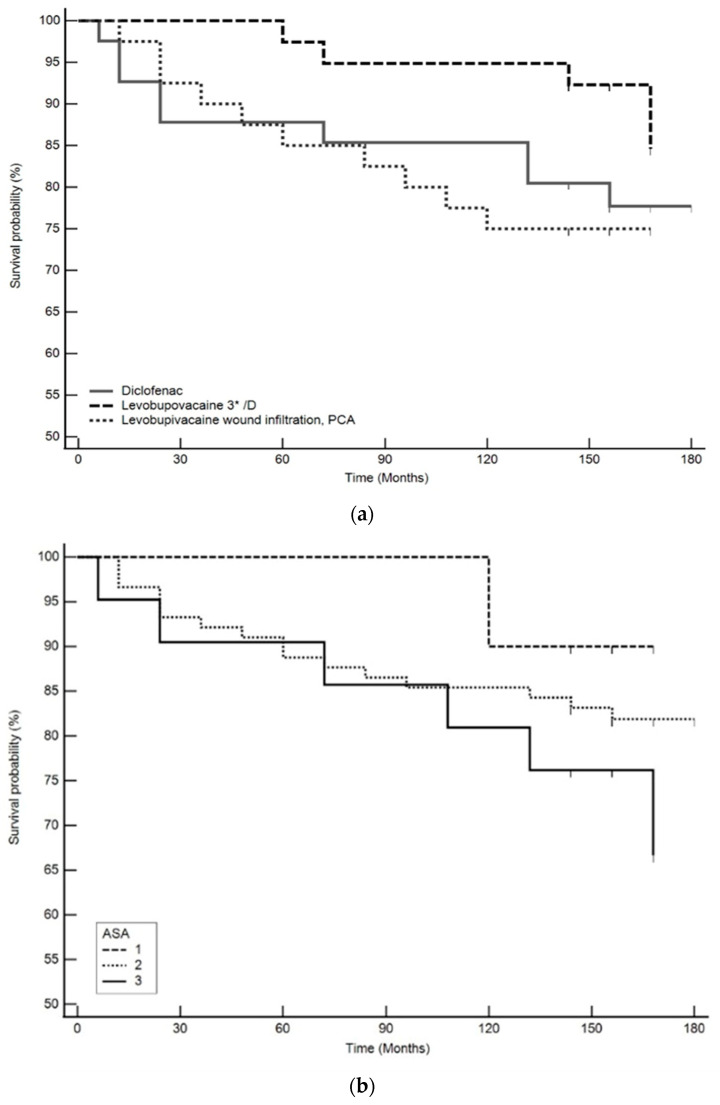
(**a**) Probability of survival in relation to the type of postoperative analgesia applied. (**b**) Probability of survival in relation to the ASA (American Society of Anesthesiologists) status at diagnosis.

**Table 1 pharmaceutics-15-02183-t001:** Demographic data of the patients undergoing breast cancer surgery with three types of postoperative analgesia at the time of surgery.

Parameters	Diclofenac (N = 41)	Levobupivacaine Bolus (N = 39)	Levobupivacaine PCA (N = 40)	*P* ^†^
Age (yr)	57 (49.5–65.5)	56 (52–65)	55.5 (49.5–60.5)	0.522
Body mass index (kg/m^2^)	29 (23–31.4)	27.8 (24.8–30.6)	26 (22.7–30.8)	0.639
ASA status	2 (2–2)	2 (2–2)	2 (2–2)	0.716
Hand grip strength (bar)	0.38 (0.34–0.44)	0.37 (0.31–0.44)	0.40 (0.34–0.47)	0.452
Stage of disease	4 (2–6)	4 (2.5–4.5)	4.5 (2.75–6.25)	0.214

**^†^** Kruskal-Wallis’s Test. ASA American Society of Anesthesiologists.

**Table 2 pharmaceutics-15-02183-t002:** Bivariate and multivariate regression analysis of factors influencing a 5-year survival.

Five-Year Survival Bivariate Regression Analysis	ß	Wald	*P* Value	HR	95% CI
Age at diagnosis	0.04	1.35	0.25	104	0.97 do 1.11
BMI	−0.01	0.04	0.84	0.99	0.88 do 1.11
ASA	0.38	0.41	0.52	1.46	0.46 do 4.65
Hand grip strength	−6.92	4.89	**0.03**	0.001	0.001 do 0.45
Stage of disease	0.28	2.62	0.11	1.33	0.94 do 1.86
Groups (diclofenac *)					
Levobupivacaine bolus	−1.63	2.21	0.14	0.20	0.02 do 1.68
Levobupivacaine PCA	−0.005	0	0.99	0.99	0.29 do 3.44
Drugs (diclofenac *)					
Any levobupivacaine group	−0.52	0.75	0.39	0.59	0.18 do 1.94
**Multivariate regression analysis**					
Hand grip strength	−8.964	5.81	**0.02**	0.0002	0 do 0.20
*Constant*	0.44	4.45	**0.03**	1.56	1.03 do 2.36

Diclofenac * was used as a comparator to other groups. BMI body mass index, ASA American Society of Anesthesiologists, PCA—patient-controlled analgesia. β coefficient—the change in the mean response. Statistically significant effects on a 5-year survival are bolded.

**Table 3 pharmaceutics-15-02183-t003:** Current survival and disease status in all patients after breast cancer surgery.

	Time from Surgery to Outcome			Diseases Stage at Diagnosis		
	n	Median (IQR)	*P **	n	Median (IQR)	*P **
Current status						
Died	23	60 (24–120)	<0.001 ^†^	23	5 (3–7)	0.03 ^‡^
Active disease, current treatment	10	156 (155–168) (min 144–max 168)		10	6 (4–7)	
No signs of disease	87	156 (156–168) (min 144–max 180)		87	4 (2–5)	

* Kruskal–Wallis test (post hoc Conover) ^†^ at the *P* < 0.05 level, the time from surgery to outcome is significantly shorter in those who died compared to those with active disease or without signs of disease. ^‡^ at the level of *P* < 0.05, the prevalence is significantly higher in patients with active disease compared to those without signs of disease.

## Data Availability

All results and data from this study are attached as Appendix A. The original data sheet is in a separate document.

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
