# Peer review of "Long-Term Outcomes of Breast Cancer Patients Receiving Levobupivacaine Wound Infiltration or Diclofenac for Postoperative Pain Relief"

_pharmaceutics, 2023, doi:10.3390/pharmaceutics15092183_

Round 1
Reviewer 1 Report (New Reviewer)
This randomized clinical trial is interesting and suitable for publication in this scientific journal. However, authors must attend to the following points:
- Add "randomized clinical trial" to the title.
- Indicate the randomization method used in the methodology and its details.
- Define in the methodology if this study was parallel or crossed.
- Clearly define the selection criteria for the sample studied.
- Explain how the sample size was calculated?
- Was it a double blind study? Who knew and identified the study treatments?
- Was the patient blinded? Explain the details.
- Which members of the research team were blinded?
- Add a studio flow chart.
-Add a figure that allows observing the means and standard deviations of pain intensity over time.
- If any, explain how losses to follow-up were managed.
- Explain the advantages and disadvantages, or strengths and weaknesses of this study.
- Add the study's public registration number on a web platform.
Author Response
Dear Reviewer 1.
Thank you for your comments. Our detailed answers are in the file attached.

Reviewer 2 Report (New Reviewer)
In abstract, “Although the best survival after ~~, statistical significance was not reached”. Why did authors say the best without statistical significance?
In Table 1: What is PCA? Define it in the legend.
Fig. 2: Make letters larger since they are too small to recognize. What is ASA? Define it in the legend.
In Table S1: What is PCA? Define in the legend. Delete “p<0.05 was considered statistically significant”.
In abstract, “Although the best survival after ~~, statistical significance was not reached”. Why did authors say the best without statistical significance?
In Table 1: What is PCA? Define it in the legend.
Fig. 2: Make letters larger since they are too small to recognize. What is ASA? Define it in the legend.
In Table S1: What is PCA? Define in the legend. Delete “p<0.05 was considered statistically significant”.
Author Response
Dear Reviewer 2
Thank you for your comments. Our reply is in the file attached.

This manuscript is a resubmission of an earlier submission. The following is a list of the peer review reports and author responses from that submission.
Round 1
Reviewer 1 Report
I have read with interest the manuscript “Effects of Postoperative Wound Infiltration Analgesia with Bupivacaine or Diclofenac on Long Term Outcomes in Patients After Breast Cancer Surgery”. In this paper the authors address the long term consequences of breast cancer surgery on chronic pain and shoulder disability, and they suggest an action of local anesthetics on cancer recurrence.
This is a prospective randomized study carefully designed and executed in 120 patients.
The authors have focused mainly on survival, which is unfortunate since their population was obviously too small to reach conclusive results in that domain. Also, quite a number of studies have hinted at an efficacy of NSAIDs to reduce breast cancer relapse (i.e. Retsky et al, Breast Cancer Res Treat. 2012 Jul;134(2):881-8). So, the choice of this comparator was perhaps not the best to study long term cancer recurrence, specifically since they could not design sub-groups of cancer types.
It would be better, especially in the discussion, to focus on the interesting results on quality of life and chronic pain, with analysis of the relevant literature and also demonstrate whether these results are due only to levobupivacaine or are a class effect extending to other local anesthetics
Author Response
REVIEWER: This is a prospective randomized study carefully designed and executed in 120 patients. The authors have focused mainly on survival, which is unfortunate since their population was obviously too small to reach conclusive results in that domain. Also, quite a number of studies have hinted at an efficacy of NSAIDs to reduce breast cancer relapse (i.e. Retsky et al, Breast Cancer Res Treat. 2012 Jul;134(2):881-8). So, the choice of this comparator was perhaps not the best to study long term cancer recurrence, specifically since they could not design sub-groups of cancer types.
When the study was created, we planned one-year follow-up. We confirmed that shoulder pain was reported in a smaller number of typical situations in groups with local anesthetic infiltration. Our first goal was to examine whether it is possible to achieve good postoperative analgesia by infiltrating the surgical area with a local anesthetic. Since the study was not published immediately, and in the meantime, other, mostly preclinical studies were done, which linked better survival with the use of local anesthetics in the perioperative period, we wanted to see if there was a difference in survival between groups of our patients. Therefore, we conducted this retrospective study. Studies examining long-term cancer recurrence included prolonged NSAID therapy. Given the suspicions about the connection between NSAIDs and cardiovascular adverse events, we did not want to use diclofenac for a longer period of time. Patients were mostly switched to paracetamol immediately after the study, before being discharged home, except for those who needed a stronger analgesic.
In the paper you mentioned (Retsky et al, Breast Cancer Res Treat. 2012 Jul;134(2):881-8) there’s a citation of another study (Forget et al. Do Intraoperative Analgesics Influence Breast Cancer Recurrence After Mastectomy? A Retrospective Analysis. Anesthesia & Analgesia 110(6):p 1630-1635, June 2010. | DOI: 10.1213/ANE.0b013e3181d2ad07). The authors associated survival with preoperatively administered ketorolac, but patients in the ketorolac group were significantly younger, and patients in other groups who had worse outcomes had significantly higher lymph node invasion and worse histologic grade. I did not find other studies that clearly demonstrated better cancer survival with perioperative NSAIDs.
REVIEWER: It would be better, especially in the discussion, to focus on the interesting results on quality of life and chronic pain, with analysis of the relevant literature, and also demonstrate whether these results are due only to levobupivacaine or are a class effect extending to other local anesthetics.
Thank you for your suggestion. There are a number of studies on the effect of levobupivacaine, but also of other local anesthetics related to the antiproliferative effect. Therefore, it is possible to conclude that the observed effect is a class effect that can be extended to other local anesthetics. We have included these comments with new references in the revised version of the manuscript.
Reviewer 2 Report
The authors present an interesting study analyzing the role of levobupivacaine (with different ways of administration) in pain control and long-term survival after breast cancer surgery.
The paper is well written and I agree with results presented about analgesic management.
I have some concerns about small sample size (as the authors report among papers limitations); It is not clear for me which is the primary outcome of the study ( improvement in long-term survival with levobupivacaine?) and how the sample size was calculated. I would ask to the authors explanations on this point. Thank you
Author Response
Reviewer 2: The authors present an interesting study analyzing the role of levobupivacaine (with different ways of administration) in pain control and long-term survival after breast cancer surgery. The paper is well written and I agree with results presented about analgesic management.
I have some concerns about small sample size (as the authors report among papers limitations); It is not clear for me which is the primary outcome of the study ( improvement in long-term survival with levobupivacaine?) and how the sample size was calculated. I would ask to the authors explanations on this point. Thank you
Dear Reviewer
The primary goal of this study was to achieve satisfactory postoperative analgesia by local anesthetic infiltration using two methods of analgesia after breast cancer surgery that was new at the time of the study. We intended to monitor analgesia in the period after surgery and at the follow-up after one year. We expected that analgesia should be better with infiltration analgesia than with diclofenac. As you can see in the supplemental Table 1, the pain was well controlled in all groups. The pain was below 4 and well-controlled in most of the patients. We also expected that infiltration with local anesthetic could lead to significant motor weakness, and therefore we included hand grip strength among the measurements. We assumed that the hand grip could be significantly weaker. Although motor weakness is transient, in the postoperative course it could interfere with independent functioning, such as personal hygiene after surgery. Based on the expected effect (f = 0.287) in the difference of numerical variables between three independent groups of subjects), we determined the sample size. We stated this in the corrected version of the text.
Given that the study was not published immediately after completion, and that in the meantime several studies were conducted that linked local anesthetics and cancer recurrence, we wanted to see if we could see a difference between the groups in our sample. Therefore, we asked the Ethics Committee to approve a retrospective study, in order to see if the use of local anesthetics has any association with the outcome.
In this revised version, we have made additional language corrections, courtesy of Mr. Bulovic, UNSW Sydney, Australia, to whom we sincerely thank.
Round 2
Reviewer 2 Report
I would like to thank the authors for their answer.
No other issues to report on my side.
Author Response
Dear Editor
Thank you for your report. We're happy that you think our manuscript is OK now. On behalf of all coauthors
Corresponding author
Slavica Kvolik